# Dominant Stickler Syndrome

**DOI:** 10.3390/genes13061089

**Published:** 2022-06-18

**Authors:** Zack Soh, Allan J. Richards, Annie McNinch, Philip Alexander, Howard Martin, Martin P. Snead

**Affiliations:** 1John van Geest Centre for Brain Repair, Vitreoretinal Research Group, University of Cambridge, Forvie Site, Robinson Way, Cambridge CB2 0PY, UK; zgys2@cam.ac.uk (Z.S.); ar204@medschl.cam.ac.uk (A.J.R.); am789@cam.ac.uk (A.M.); hm208@cam.ac.uk (H.M.); 2NHS England Highly Specialised Stickler Syndrome Diagnostic Service, Cambridge University, NHS Foundation Trust, Addenbrooke’s Hospital, Hills Road, Cambridge CB2 0QQ, UK; philip.alexander@addenbrookes.nhs.uk

**Keywords:** Stickler syndrome, retinal detachment, vitreous, giant retinal tear, *COL2A1*, *COL11A1*

## Abstract

The Stickler syndromes are a group of genetic connective tissue disorders associated with an increased risk of rhegmatogenous retinal detachment, deafness, cleft palate, and premature arthritis. This review article focuses on the molecular genetics of the autosomal dominant forms of the disease. Pathogenic variants in *COL2A1* causing Stickler syndrome usually result in haploinsufficiency of the protein, whereas pathogenic variants of type XI collagen more usually exert dominant negative effects. The severity of the disease phenotype is thus dependent on the location and nature of the mutation, as well as the normal developmental role of the respective protein.

## 1. Introduction

Stickler syndrome is a hereditary connective tissue disorder that is usually caused by pathogenic variants of genes encoding for fibrillar collagens II, IX, and XI, which are found in the vitreous humour as well as hyaline and elastic cartilage. It affects 1 in 7500 to 9000 newborns [1], and it is the leading cause of heritable rhegmatogenous retinal detachment and rhegmatogenous retinal detachment in childhood [2].

## 2. Historical Overview

Stickler syndrome was first reported by Gunnar Stickler et al. in 1965 as “hereditary progressive arthro-opthalmopathy” [3] and was initially thought to be an autosomal dominant monogenic condition affecting both the joints and the eyes. Two years later, it was reported that the mother and proband also suffered from sensorineural hearing loss. Linkage analysis implicated the *COL2A1* gene for type II collagen in a majority of cases [4], and the first pathogenic variant of the gene was subsequently confirmed by Ahmad et al. [5]. Molecular genetic analysis subsequently also confirmed a *COL2A1* c.1222–2A > G pathogenic variant in the original pedigree reported by Gunnar Stickler [6].

Snead et al. were the first to report different vitreous phenotypes linked to genetic heterogeneity in Stickler syndrome [7,8,9], and this was confirmed with the first report of a pathogenic variant of *COL11A1* in type 2 Stickler syndrome [8] and *COL11A2* in non-ocular type 3 Stickler syndrome (otherwise known as dominant otospondylomegaepiphyseal dysplasia (OSMED)) [10,11].

A classification of the subgroups of Stickler syndrome based on vitreous phenotypes has developed over the last 20 years and has proven to be a reliable method of both diagnosing Stickler syndrome and guiding molecular analysis [12]. More recently, cases of Stickler syndrome associated with pathogenic variants in an increasing number of gene loci have been reported, resulting in the identification of at least 10 different subgroups of Stickler syndrome associated with pathogenic variants in 8 distinct genes (with 5 of these subgroups falling under the category of autosomal dominant) (Table 1) [2,13]. This article reviews the autosomal dominant forms of Stickler syndrome, which account for the majority of patients that present to clinicians.

## 3. Clinical Features

Most subgroups of dominant Stickler syndrome are caused by pathogenic variants of the genes coding for types II, IX, and XI collagen, which are principally found in vitreous humour [14] and articular and hyaline cartilage. Patients may exhibit varying combinations of ocular, auditory, musculoskeletal, and orofacial abnormalities. The secondary vitreous humour completes embryological development by 8–14 weeks of intrauterine growth, and abnormalities in embryological development of vitreous humour can be readily visualised using slit-lamp bio-microscopy, greatly assisting clinical diagnosis (see Figure 1) as well as differentiating between subgroups (types 1, 2, 7, and ocular-only forms (vitreous anomaly) from type 3 (non-ocular, normal vitreous phenotype)). Vitreous phenotyping is key to the diagnosis of ocular-only subgroups of Stickler syndrome, as systemic features are usually mild or absent in such patients [12].

The diagnosis of Stickler syndrome should be considered in any of the following [13]:  (i)Infants with a history of congenital myopia in association with deafness (ii)Infants born with cleft palate or Pierre Robin Sequence in association with myopia(iii)Infants with joint hypermobility and/or epiphyseal dysplasia in association with myopia(iv)Individuals suffering rhegmatogenous retinal detachment with a family history of rhegmatogenous retinal detachment.

## 4. Differential Diagnosis

Patients with other connective tissue disorders may present with similar symptoms and features to Stickler syndrome (including retinal detachment) and should be considered as part of the differential diagnosis. The locations of the genes involved in AD Stickler syndrome and the differential diagnoses are shown in Figure 2.

Marfan syndrome is caused by pathogenic variants of the gene encoding fibrillin-1 and may present with abnormalities in the cardiovascular, skeletal, ocular, pulmonary, integumentary, and central nervous systems [15]. Patients with Marfan syndrome may have a tall stature, long extremities, arachnodactyly, kyphoscoliosis, joint hypermobility, and a crowded or high arched palate [15]. Occasionally, patients with AD Stickler syndrome may exhibit some musculoskeletal features suggestive of a “Marfanoid” habitus, including slender extremities and joint laxity, both of which are also recognised in many allied connective tissue disorders (see below).

Loeys–Dietz syndrome is an autosomal dominant condition characterised by hypertelorism, bifid uvula/cleft palate, and aortic aneurysm with tortuosity [16]. It is caused by pathogenic variants affecting the proteins involved in the TGFβ signalling pathway (including *TGFBR1*, *TGFBR2*, *SMAD3,* and *TGFB2*), resulting in increased TGFβ signalling. The systemic nature of the disease means patients may also present with skeletal, craniofacial, integumentary, and ocular manifestations. Patients may thus exhibit “Marfanoid” features such as arachnodactyly and joint hypermobility [17].

Wagner syndrome is an autosomal dominant vitreoretinopathy characterised by accelerated, premature cataract (“catarata complicata”), chorioretinal atrophy, and increased risk of retinal detachment [18]. On slit-lamp examination, patients demonstrate a hypoplastic vitreous with extensive pre-retinal membrane [18]. Myopia is usually less severe than in Stickler syndrome, and the risk of rhegmatogenous retinal detachment appears to be less than Stickler syndrome. Wagner syndrome is caused by pathogenic variants of the *VCAN* gene encoding Versican, which is expressed in many soft tissues including vitreous. Pathogenic variants affect the splicing of exon 8, which is utilised in the V0 and V1 isoforms of Versican, resulting in the overrepresentation of the V2 and V3 isoforms [19].

Czech dysplasia metatarsal type is an autosomal dominant disorder with the hallmark of shortened third and fourth toes caused by metatarsal hypoplasia [20]. Patients also exhibit progressive spondyloarthropathy, but orofacial abnormalities have not been reported. This disorder is exclusively caused by the p.(Arg475Cys) pathogenic variant within the triple helix region of the *COL2A1* gene, which has a dominant negative effect and is thought to negatively affect the local integrity of the α1 chain of type II collagen [20].

Knobloch syndrome is an autosomal recessive disorder associated with pathogenic variants of the *COL18A1* gene encoding the α1 chain of type 18 collagen [21]. As type 18 collagen is involved in the development of the eye, patients often present with ocular abnormalities such as retinal detachment, high myopia, featureless iris crypts, congenital cataracts, lens subluxation, and severe RPE atrophy resulting in prominent choroidal vasculature [21]. Extraocular manifestations include occipital encephalocoele, bifid ureters, dental caries, and hypermobile joints.

Patients with autosomal dominant rhegmatogenous retinal detachment suffer from increased risk of rhegmatogenous retinal detachment but differ from Stickler syndrome patients, as they have no systemic features and do not demonstrate membranous or beaded vitreous anomalies [22]. A family with this condition was also found to have a pathogenic variant in *COL2A1* p.(Gly318Arg) [22], and the same gene has also been found to have a weak association with an increased risk of rhegmatogenous retinal detachment in the general population [23].

Donnai–Barrow syndrome is an autosomal recessive condition that may also present with severe myopia and an increased risk of retinal detachment, but the associated congenital diaphragmatic hernia, congenital heart defects, corpus callosum agenesis, hypertelorism, and sensorineural hearing loss should facilitate the differential diagnosis from Stickler syndrome [24]. Patients may also develop severe myopia and have increased risk of retinal detachment. The disease is caused by pathogenic variants of the *LRP2* gene that codes for megalin, which is a transmembrane receptor responsible for the re-uptake of many ligands crucial for embryonic development. Disruptions in interactions between megalin and sonic hedgehog may also contribute to abnormal brain and lung development [25].

The Ehlers–Danlos group of connective tissue disorders may also share joint hypermobility and myopia with Stickler syndrome, but they are associated with skin hyperextensibility and tissue fragility [26]. The underlying pathogenic variant usually affects the synthesis and processing of different types of collagen. The most common subtypes of Ehlers–Danlos syndrome are the hypermobile, classic, and vascular types, with pathogenic variants of *COL5A1* and *COL5A2* causing the classic subtype, and pathogenic variants of *COL3A1* causing the vascular subtype. Although *COL5A2* is expressed in the eye, the vitreous phenotype in these patients is compatible with their refractive error.

## 5. Genotype and Phenotype

### 5.1. Type 1 AD Stickler Syndrome

Type 1 Stickler syndrome occurs in more than 80% of patients with Stickler syndrome [27]. The *COL2A1* gene implicated in type 1 Stickler syndrome expresses the α1 chain of type II collagen. Type II collagen is the major structural constituent of composite fibrillar collagen molecules found in cartilage and vitreous, which consists of a core of type XI collagen surrounded by type II collagen. Type IX collagen is one of the molecules that connects these collagen fibres to other components of the extracellular matrix. Trivalent cross-links connect type II collagen molecules with one another in a head-to-tail configuration, whereas divalent cross-links bind type XI collagen and type IX collagen with type II collagen molecules [28]. The structure of the composite fibril is shown in Figure 3. There is thus some phenotypic overlap between defects in these types of collagen, as they are conformationally associated with each other. This aligns with the fact that different subgroups of Stickler syndrome can be caused by pathogenic variants coding for types II, IX, and XI collagen, with similar systemic manifestations. Notably, the α(1)II chain encoded by *COL2A1* is also a minor component of type XI collagen. In vitreous, the α(2)XI chain in type XI collagen is replaced by α(2)V chain of type V collagen in a proportion of molecules, but pathogenic changes in *COL5A2* have not been reported to result in an abnormal vitreous.

#### 5.1.1. Nonsense-Mediated Decay (Haploinsufficiency)

Pathogenic variants of the *COL2A1* gene that typically cause type 1 Stickler syndrome are usually premature stop codons or frameshift mutations that lead to nonsense-mediated decay of the mRNA transcript, resulting in haploinsufficiency of type II collagen [29]. Pathogenic variants at splice sites can also result in type 1 Stickler syndrome via the creation of cryptic splice sites and subsequently cause a shift in the reading frame, ultimately resulting in haploinsufficiency through nonsense-mediated decay [29]. Similarly, apparently silent mutations in *COL2A1* have also been shown to lead to haploinsufficiency by creating a de novo donor splice site within an exon, causing a shift in the reading frame of the differentially spliced mRNA transcript [30].

In contrast, pathogenic variants that produce an in-frame mRNA transcript through missense mutations or nonsense-associated altered splicing and thus exon skipping result in full length mRNA transcripts that are usually able to evade nonsense-mediated decay. The mutant mRNA transcript is then translated, and the mutant protein associates with two other α1 chains to form an abnormal trimeric collagen molecule. As type II collagen is homotrimeric, in a patient with a heterozygous mutation, 7/8 molecules will contain at least one mutant α1 chain. Instead of haploinsufficiency, the pathogenic variant has a dominant negative effect, which results in a more severe phenotype usually seen in conditions such as Kniest dysplasia and Spondyloepiphyseal Dysplasia Congenita (SEDC). This is illustrated in Table 2. As a result, pathogenic variants that result in an in-frame mRNA transcript are rarer in patients with Stickler syndrome [31]. Differences in splicing factors between cell types and individuals may affect the proportion of mRNA transcripts that are targeted by the nonsense-associated altered splicing pathway, which may explain why the resulting phenotype in some patients is the milder Stickler syndrome rather than Kniest dysplasia and may also explain the variable systemic phenotype seen in patients with type 1 Stickler syndrome.

#### 5.1.2. Effects on Phenotype

It has been proposed that a critical mass of type II collagen is required for the secondary vitreous to develop in the embryo, and haploinsufficiency results in only a vestigial vitreous gel forming in the retrolental space, giving rise to the type 1 membranous vitreous phenotype [32]. It is unknown if haploinsufficiency of the minor component of type XI collagen encoded by *COL2A1* has any additional effect on vitreous lamellar morphology [32]. Patients with type 1 Stickler syndrome have a higher risk of developing giant retinal tears, which can subsequently develop into retinal detachment (see Figure 4). 

Type II collagen is a major component of cartilage, and any defects in type II collagen affect the structural integrity of tissues containing cartilage. It is likely that haploinsufficiency of type II collagen during development results in the inability to form a normal meshwork in cartilage and the delayed development of skeletal structures, causing congenital musculoskeletal and orofacial abnormalities such as cleft palate and spondyloepiphyseal dysplasia. Patients also present with failure to fuse growth plates, which could be caused by abnormal distribution of type II collagen in the physis [2]. Type II collagen is also present in inner ear structures, the tympanic membrane, and the joints between the bones of the middle ear, and it is responsible for the shape of the tympanic membrane [33]. Associated craniofacial abnormalities may also increase the risk of otitis media in patients, which can lead to subsequent hearing loss [34]. Palate abnormalities may also disrupt proper Eustachian tube function, increasing the risk of hearing loss [35]. These factors could explain the prevalence of tympanic membrane hypermobility and hearing loss in patients with type 1 Stickler syndrome.

### 5.2. Ocular-Only AD Stickler

The majority of pathogenic variants that cause the ocular-only variant of Stickler syndrome are missense and nonsense mutations that affect exon 2 of *COL2A1* [36]. Exon 2 of *COL2A1* has been found to be alternatively spliced, and the long form of type II collagen (type IIA isoform) is expressed in chondrogenic tissue during development and in the vitreous, whereas the short form (type IIB isoform) is predominantly expressed in adult cartilage. Missense and nonsense mutations in exon 2 thus only affect type IIA collagen (principally) in the vitreous, giving rise to the ocular-only variant of Stickler syndrome [36].

The ocular-only phenotype of Stickler syndrome is not wholly confined to pathogenic variants of exon 2 of *COL2A1*. Pathogenic variants in other regions of *COL2A1* can cause tissue-specific missplicing, leading to variable systemic effects, and the rapid completion of vitreous embryogenesis compared to cartilage may make it more susceptible to falls in the level of type II collagen, resulting in a predominantly ocular-only phenotype [29].

### 5.3. Type 2 AD Stickler Syndrome

Type 2 Stickler syndrome is caused by pathogenic variants of the *COL11A1* gene, which codes for the α1 chain of type XI collagen. The α1 chain associates with the α2 chain encoded by *COL11A2* and the α(1)II chain encoded by *COL2A1* to form the heterotrimeric type XI collagen.

#### 5.3.1. Exon Skipping and Missense Mutations (Dominant Negative Effect)

Unlike pathogenic variants of *COL2A1*, pathogenic variants of *COL11A1* tend to predominantly affect splice sites, resulting in exon skipping and the production of an in-frame mRNA transcript. In particular, the donor splice site of exon 50 has been identified as a mutational hotspot [37]. Mutant mRNA transcripts thus do not usually undergo nonsense-mediated decay and are normally translated and associated with other α chains to form a mutant type XI collagen trimer. This results in a dominant negative effect instead of haploinsufficiency and is associated with more severe disease phenotype. Missense mutations involving the substitution of the obligate glycine in the Gly-X-Y helix have also been associated with type 2 Stickler syndrome, although they may not be dominant as heterozygous carriers presented with a mild phenotype [38]. Pathogenic variants of *COL11A1* resulting in haploinsufficiency are correlated with a milder, less obvious disease phenotype. As such, nonsense mutations that produce a premature stop codon are rare in patients.

#### 5.3.2. Effects on Phenotype

Type XI collagen is a minor constituent of the collagen fibre, so bulk formation of the vitreous can proceed despite pathogenic variants of *COL11A1*. However, type XI collagen has the role of regulating the diameter and spacing of type II collagen fibres during collagen fibrillinogenesis [39], so pathogenic variants of *COL11A1* can result in the formation of collagen fibres with irregular diameter. This may explain why patients with type 2 Stickler syndrome exhibit a beaded vitreous phenotype, where the vitreous lamellae composed of collagen is irregular in size and often thickened [32].

Since type XI collagen is responsible for regulating the nucleation and lateral growth of collagen fibrils, the lack of type XI collagen may result in a lack of collagen fibrils in cartilage, producing the craniofacial and musculoskeletal abnormalities characteristic of type 2 Stickler syndrome. Hearing loss is also more common and more severe in patients with type 2 Stickler syndrome compared to type 1 Stickler syndrome, with 82.5% of patients found to have hearing loss compared to a prevalence of 52% in type 1 Stickler syndrome patients [40]. *COL11A1* and *COL11A2* have been found to be expressed in the developing cochlea and are thus likely to be implicated in the development of the cochlea [41]. As no macro-deformities in the inner ear have been observed in patients with type 2 Stickler syndrome, it has been suggested that microstructure irregularity is responsible for hearing loss in these patients, particularly the mechanical properties of the cochlear partition [12]. *COL11A1* collagen fibrils are also present in the fibrous layer of the tympanic membrane, so pathogenic variants of *COL11A1* may cause tympanic membrane hypermobility seen in patients [42].

### 5.4. Type 3 AD Stickler Syndrome (Non-Ocular)

Type 3 Stickler syndrome is caused by pathogenic variants of the *COL11A2* gene, which codes for the α2 chain of collagen XI. Unlike the α1 chain, *COL11A2* is mostly expressed in cartilage instead of in the vitreous (where it is replaced by the α(2)V chain of type V collagen), resulting in a disease phenotype that does not affect the eye. As such, type 3 Stickler syndrome is also known as the non-ocular variant, or autosomal dominant otospondylomegaepiphyseal dysplasia (OSMED, MIM 184,840).

#### 5.4.1. Exon Skipping and Missense Mutations (Dominant Negative Effect)

Pathogenic variants of *COL11A2* usually result in a dominant negative effect with more severe disease phenotype. These include missense mutations within the helical domain of collagen that may destabilise the helix. Seemingly silent mutations can also affect the splicing of the mRNA transcript, through the creation of either a donor splice site or an acceptor splice site [38]. Several in-frame deletions have also been reported to cause type 3 Stickler syndrome [11,43]. Pathogenic variants affecting exonic splicing elements may favour the production of in-frame mRNA transcripts through nonsense-associated altered splicing [44]. The dominant negative effect of these pathogenic variants stem from the ability of the mutant protein to associate with other α chains, causing 7/8 of type XI collagen heterotrimers to be dysfunctional.

#### 5.4.2. Effects on Phenotype

Similar to type 2 Stickler syndrome, the lack of type XI collagen may result in lower levels of collagen in cartilage matrix. According to zebrafish models, this may result in disorganised patterns of collagen, decreased joint space, and articular cartilage degradation [45]. These factors may predispose patients to systemic abnormalities such as premature osteoarthritis. *COL11A2* is also expressed in the tectorial membrane of the cochlea and interacts with other tectorial components [41]. Pathogenic variants of *COL11A2* may cause abnormal collagen distribution in the tectorial membrane and lead to non-progressive childhood-onset hearing loss, which is present in 94.1% of patients [40].

### 5.5. Type 7 AD Stickler Syndrome

The p.(Gly44Ter) pathogenic variant in the *BMP4* gene was linked to Stickler syndrome phenotype in one case report, and it is the only non-collagenous form of AD Stickler syndrome reported [46]. *BMP4* is a growth factor that has an important role during embryonic development as well as ocular development, and other pathogenic variants of *BMP4* usually result in severe developmental abnormalities such as anophthalmia/microphthalmia, structural brain anomalies, and syndactyly [47].

#### 5.5.1. Severe Truncation (Haploinsufficiency)

The p.(Gly44Ter) pathogenic variant is expected to create a premature stop codon in the N-terminal prodomain of *BMP4*, causing a severely truncated protein to be expressed. Although nonsense-mediated decay was not observed, the severely truncated protein is unlikely to be able to interact with other growth factors, and so haploinsufficiency instead of a dominant negative effect is the likely result. More downstream mutations found in other diseases were associated with a different ocular and systemic phenotype, including anophthalmia/microphthalmia and congenital glaucoma, instead of megalophthalmos associated with Stickler syndrome.

#### 5.5.2. Effects on Phenotype

During development, levels of BMP4 are crucial in determining the dorsal–ventral axis of the eye [48]. It is thus possible that haploinsufficiency of *BMP4* results in increased axial length of the eyeball, explaining the congenital myopia in type 7 Stickler syndrome. BMP4 also aids in the development of the lens, although no lens abnormalities were reported in patients. Abnormal levels of BMP4 may affect its interactions with TGFβ1, which may result in the increased risk of retinal detachment in patients. Due to the structural similarity of BMP2 and BMP4, BMP4 may also interact with the cysteine rich domain of type II collagen encoded by *COL2A1*. As pathogenic variants in this region of *COL2A1* can result in type 1 Stickler syndrome, it has been suggested that haploinsufficiency of *BMP4* may affect these interactions with type II collagen, disrupting the development of vitreous and cartilage, producing a similar phenotype. This includes features such as congenital hypoplasia of vitreous, sensorineural hearing loss, high-arched palate, and retrognathia. 

BMP4 is an important promoter and regulator in the development of the kidneys and urinary tract. As such, congenital anomalies of the kidney and urinary tract (CAKUT) have previously been associated with pathogenic variants of *BMP4* [49]. In addition to developmental anomalies in vitreous and cartilage, patients with type 7 Stickler syndrome may also present with CAKUT, which is a unique clinical feature not observed in any other subgroup of Stickler syndrome and is thus an important diagnostic marker to differentiate type 7 Stickler syndrome from other subgroups of Stickler syndrome. 

## 6. Conclusions

Stickler syndrome consists of a group of hereditary collagenopathies characterised by increased risk of rhegmatogenous retinal detachment and developmental abnormalities in vitreous and/or tissues containing cartilage. Differential diagnoses for Stickler syndrome include, but are not limited to: Marfan syndrome, Loeys–Dietz syndrome, Wagner syndrome, Czech dysplasia metatarsal type, Knobloch syndrome, autosomal dominant rhegmatogenous retinal detachment, Donnai–Barrow syndrome, and Ehlers–Danlos syndrome. The subgroups of Stickler syndrome which are inherited in an autosomal dominant fashion include types 1, 2, 3, 7, and the ocular-only variant. Depending on the affected gene, pathogenic variants can either cause haploinsufficiency through:Creation of premature stop codon through nonsense or frameshift mutations (possibly through affecting splicing) leading to nonsense-mediated decay;Loss-of-function mutations that prevent the mutant protein from interacting with other proteins.

Alternatively, they can exert a dominant negative effect through:3.Missense or nucleotide adding/deleting mutations that produce an in-frame transcript.4.Nonsense mutations that lead to nonsense-associated altered splicing and thus exon skipping.

## Figures and Tables

**Figure 1 genes-13-01089-f001:**
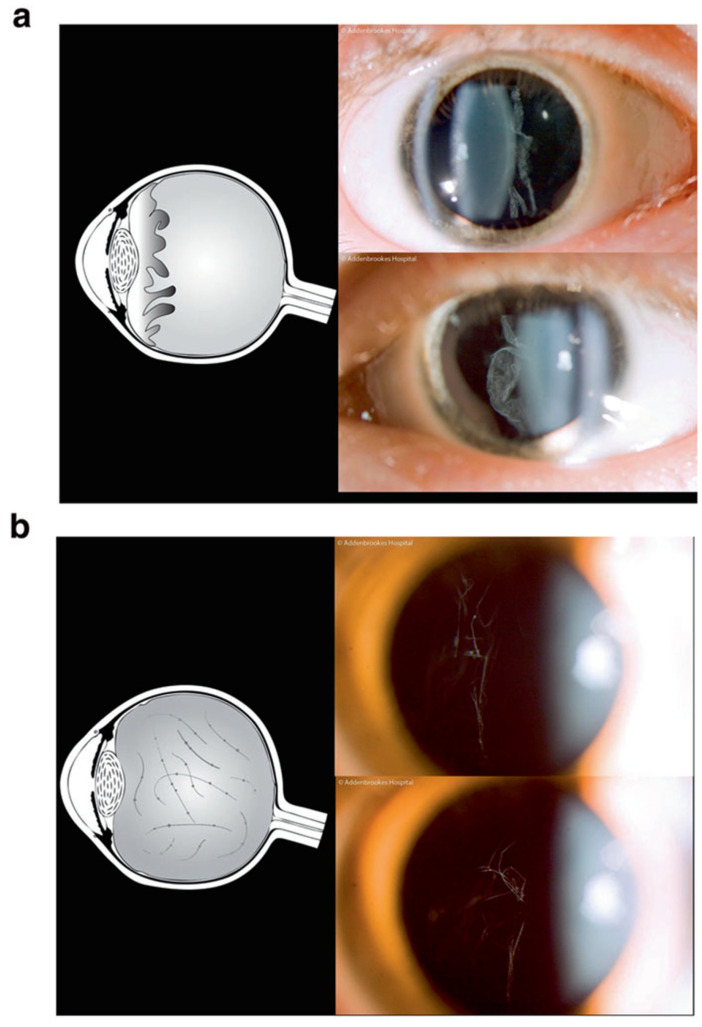
Schematic illustrations of vitreous phenotypes associated with Stickler syndrome. (**a**) Membranous congenital vitreous anomaly, (**b**) Beaded congenital vitreous anomaly. Reproduced with permission from Snead et al. [13].

**Figure 2 genes-13-01089-f002:**
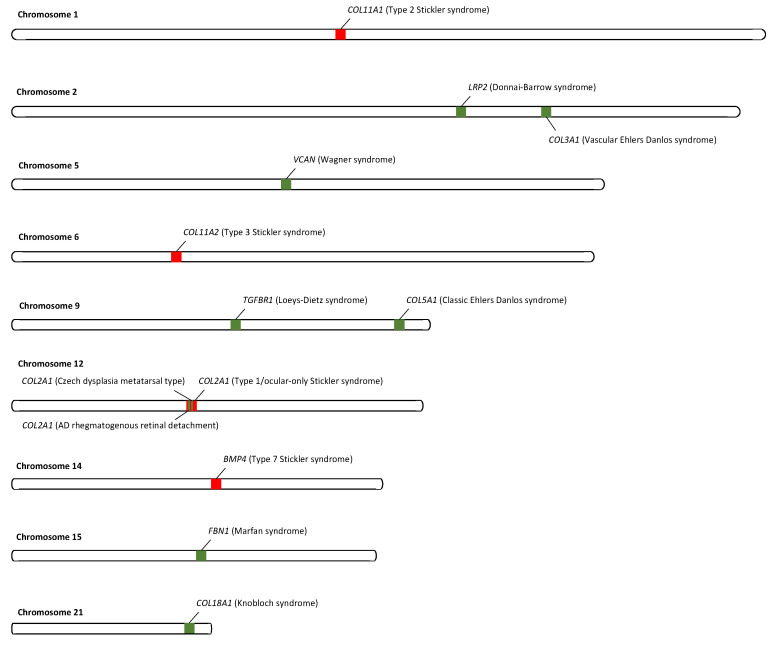
Locations of pathogenic variants associated with each syndrome. Genes associated with AD Stickler syndrome are in red, and genes associated with the differential diagnoses are in green.

**Figure 3 genes-13-01089-f003:**
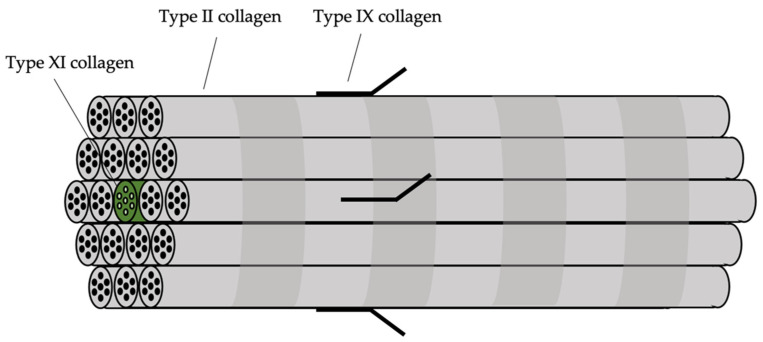
Structural relationship between types II, IX, and XI collagen.

**Figure 4 genes-13-01089-f004:**
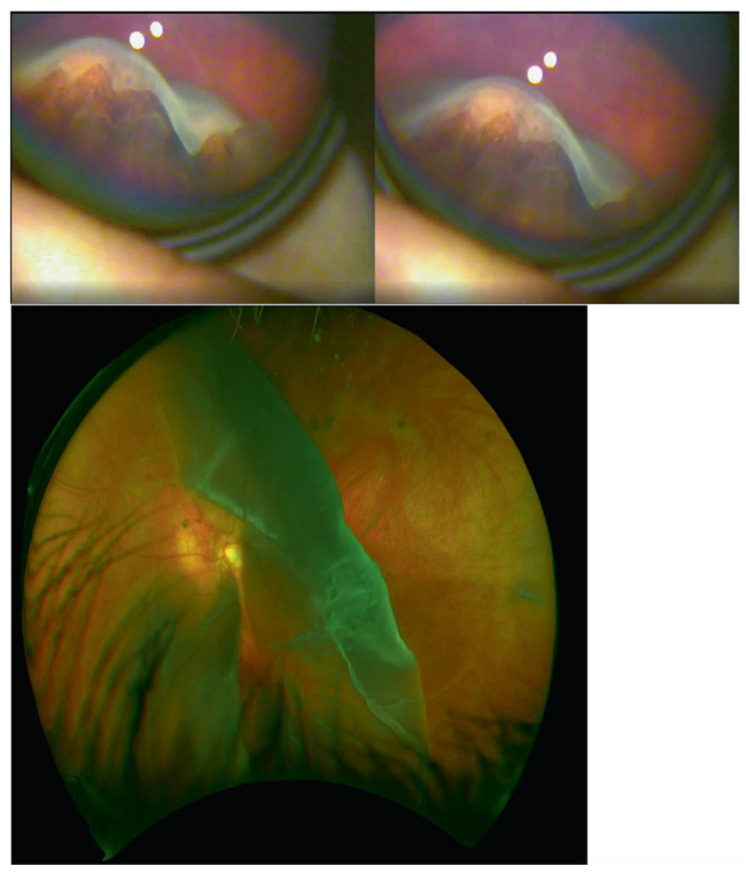
(**Top**) 210 degrees Giant Retinal Tear identified in an otherwise asymptomatic 8-year-old child with type 1 Stickler syndrome. (**Bottom**) Giant Retinal Tear in child with undiagnosed type 1 Stickler syndrome. Reproduced with permission from Snead et al. [13].

**Table 1 genes-13-01089-t001:** Subgroups of **AD Stickler** syndrome and their key clinical features. Adapted with permission from Snead et al. [13].

Subtype of AD Stickler Syndrome	Gene	Cytogenetic Location	Distinguishing Features	Phenotype MIM no.
Type 1	*COL2A1*	12q13.11	Type 1 membranous congenital vitreous anomaly, retinal detachment, congenital megalophthalmos, deafness, arthropathy, cleft palate. High risk of blindness	108,300
Ocular-only	*COL2A1*	12q13.11	Type 1 membranous congenital vitreous anomaly, retinal detachment, congenital megalophthalmos. No systemic features. High risk of blindness	609,508
Type 2	*COL11A1*	1p21.1	Beaded type 2 congenital vitreous anomaly, retinal detachment, congenital megalophthalmos, deafness, arthropathy, cleft palate.	604,841
Type 3 (dominant OSMED)	*COL11A2*	6p21.32	Non-ocular form of Stickler syndrome. Normal vitreous and ocular phenotype, deafness, arthropathy, cleft palate.	184,840
Type 7	*BMP4*	14q22.2	Hypoplastic vitreous, retinal detachment, deafness, arthropathy, palate abnormality, renal dysplasia.	To be confirmed

**Table 2 genes-13-01089-t002:** Probabilities of possible outcomes of haploinsufficiency and dominant negative pathogenic variants.

Product of Wild-Type Allele	Product of Mutated Allele	Collagen Trimer Product (Possibility 1)	Collagen Trimer Product (Possibility 2)	Phenotypic Result of the Pathogenic Variant
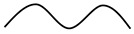	No product	50% of normal amount 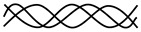		Haploinsufficiency
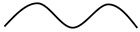	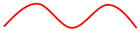	7/8 chance of abnormal trimer 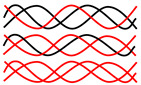	1/8 chance of normal trimer 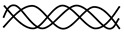	Dominant negative

## Data Availability

Data sharing not applicable.

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
