# Peer review of "Dominant Stickler Syndrome"

_genes, 2022, doi:10.3390/genes13061089_

Round 1
Reviewer 1 Report
The authors present an academic collection of genetic causes of autosomal dominant Stickler syndrome. Their focus lies on mutations and their potential mechanistic cause of the phenotype with emphasis on haploinsufficiency and dominant negative effects. They also point to differential diagnostics and summarize briefly other diseases with overlapping features. The manuscript is well written.
Following the following points would be beneficial to this review.
1. In general, references are not given, when needed. A few examples are listed below, but the authors may want to check the entire manuscript for this. After all, this is a review and references are crucial.
a. line 40 to 46 contain several important statements, which should be supported by a reference.
b. All examples of differential diagnosis need a reference at the end of the first sentence.
c. Lines 163 to 171 should be supported by references
d. Line 195 to 198 should have a reference given
2. The list of references should be checked carefully. 2 examples struck me as incomplete or wrong: Ref 3 and ref 17.
3. In the Introduction, It would be great to read how many “Stickler”genes have been found as of today.
4. Examples of differential diagnosis are well listed, but I imagine a graphic display where the different collagen genes are circled with the different syndromes associated. That would allow a clear picture at a glance.
5. Table 2 is not clear to me. Just a few examples: Does “Type of mutation” belong to the column or the row (neither is a good fit). Haploinsufficiency for Wild Type product does not fit either etc.
6. The conclusions are meager: A comment about differential diagnosis would be helpful and more complete. Further, line 356 is at an odd place.
Author Response
Dear Sir/Madam,
Thank you for your constructive comments. Please see the attachment.

Reviewer 2 Report
This is a superb paper, an excellent and comprehensive review of a common and important condition to know about.
The only (very minor) point is that "mutations" is used in several places where pathogenic variant might be more appropriate. Otherwise, no comments
Author Response
Dear Sir/Madam,
Thank you for your kind comments. We have made the appropriate changes to the phrasing of the manuscript according to your comments.
Round 2
Reviewer 1 Report
Thank you authors for your immediate response. The manuscript is now clear and informative, and the new Figure 2 helps to grasp the differential diagnosis and the different genes at a glance.
Please pay attention to the final criticism:
Reference 18 is still not correct. Please insert the correct one.
Line 44: you mentioned the number of subtypes of Stickler syndrome but what I was asking for was to include the total number of genes, known today, to be involved in AD Stickler Syndrome.
Author Response
Dear Sir/Madam,
Thank you once again for your constructive comments. The manuscript has been revised according to your feedback.
Point 1: We apologise for the error in the reference. This has now been fixed.
Point 2: Lines 44-46 have been updated to mention the number of genes associated with Stickler syndrome, as well as how many subgroups belong to the autosomal dominant category. In addition, the number of Stickler syndrome subgroups has been changed from 11 to 10 to reflect more recent literature.